# Execution-Guided Neural Program Decoding

## Abstract

We present a neural semantic parser that translates natural language questions into *executable* SQL queries with two key ideas. First, we develop an encoder-decoder model, where the decoder uses a simple type system of SQL to constraint the output prediction, and propose a value-based loss when copying from input tokens. Second, we explore using the execution semantics of SQL to repair decoded programs that result in runtime error or return empty result. We propose two model-agnostics repair approaches, an ensemble model and a local program repair, and demonstrate their effectiveness over the original model. We evaluate our model on the WikiSQL dataset and show that our model achieves close to state-of-the-art results with lesser model complexity.

## 1. Introduction

Developing effective semantic parsers to translate natural language questions into logical programs has been a long-standing goal (Poon, 2013; Zettlemoyer & Collins, 2005; Pasupat & Liang, 2015; Li et al., 2005; Gulwani & Marron, 2014). Recent work has shown that recurrent neural networks with attention and copying mechanisms (Dong & Lapata, 2016; Neelakantan et al., 2016; Jia & Liang, 2016) can be used to successfully build such parsers. Notably, Zhong et al. (2017) recently introduced the Seq2SQL model for translating questions to SQL queries using supervised learning. The model uses separate decoders for different parts of a query (i.e., aggregation operation, target column, and where predicates) and reinforcement learning to learn semantically equivalent queries beyond supervision.

In this paper, we present a new model for decoding programs that leverages one key property of programs that programs have well-defined deterministic semantics and

---
[1]Anonymous Institution, Anonymous City, Anonymous Region, Anonymous Country. Correspondence to: Anonymous Author <anon.email@domain.com>.

Preliminary work. Under review by the International Conference on Machine Learning (ICML). Do not distribute.

are executable. Our model is an extension of the sequence-to-sequence model with attention (Bahdanau et al., 2014) for natural language to SQL program translation. Figure 1 shows an example table-question pair and how our system generates the answer by executing the synthesized SQL program. There are two key ideas in our model. First, instead of designing multiple decoders (Krishnamurthy et al., 2017), we use a simple *type system* to control the decoding mode at each decoding step (cf. Sect. 2). Based on the SQL grammar, a decoder cell is specialized to either select a token from the SQL built-in vocabulary, generate a pointer over the table header and the input question to copy a table column, or generate a pointer to copy a constant from the user's question. We use a value-based loss function that transfers the distribution over the pointer locations in the input into a distribution over the set of tokens observed in the input, by summing up the probabilities of the same value appearing at different input indices. Second, we use the execution semantics of SQL to repair (partially) decoded programs that either trigger runtime error or return empty result during execution. In particular, we study two model-independent repair approaches: (1) an ensemble model approach where erroneous programs are repaired using programs generated from other models in a model ensemble and (2) a local program repair approach that repairs programs on-the-fly during the decoding process based on its partial evaluation result. Our study result shows that both strategies can effectively improve the accuracy of the base model.

We evaluate our approach on the recently released WikiSQL dataset (Zhong et al., 2017), a corpus consisting of over 80,000 natural language question and pairs. Our results in Sect. 3 show that our end-to-end model achieves a similar test accuracy (78.3% execution accuracy) to that of the state-of-the-art Coarse2Fine model (Dong & Lapata, 2018b) (78.5% execution accuracy) without needing a separate neural table encoder to encode table values or an intermediate decoder and encoder to embed the program sketch. Using a series of ablation experiments, we show that our model independent repair strategies can effectively boost base model performance (with an improvement from 71.9% to 78.3%). More importantly, the execution guided program decoding can be composed with more advanced models for further performance improvement and even to other neural program synthesis domains (Parisotto et al., 2016).

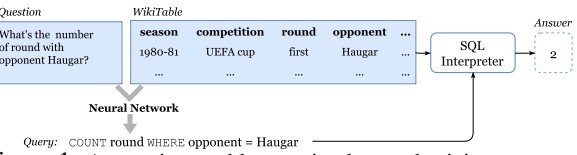

Figure 1: Answering a table question by synthesizing a query and executing it on the provided table.

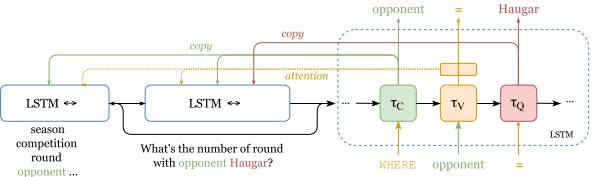

Figure 2: Overview of the base model. The model encodes table columns as well as the user question with a BiLSTM and then decodes the hidden state with a typed LSTM, where the decoding action for each cell is statically determined.

## 2. Model

In this section, we introduce the proposed framework, including the base model and the execution-guided program decoding algorithm.

### 2.1. Base model

We generate SQL queries from questions using an RNN-based encoder-decoder model with attention and copying mechanisms (Vinyals et al., 2015; Gu et al., 2016; Zhong et al., 2017). Besides, we use the known structure of SQL to *statically* determine the "type" of output of a decoding step while generating the SQL query. For example, since the grammar determines that the third token (after the aggregation function) in any query has to be a column name (i.e., the aggregated column), we only need to consider column names when decoding the hidden state at this position. To generalize this idea, we use a static type system to restrict decoder candidates for each decoding position: if the target token is a column name, we enforce the use of a copying mechanism to copy a token matching one of the table header; if the target token a constant, we restrict the copy header to copy from the user question; otherwise we project the hidden state to a built-in vocabulary to obtain a built-in SQL operator. This means that we only need to maintain a small built-in decoder vocabulary (sized 15) for all operators.

The encoder is a bidirectional LSTM, which takes the concatenation of the table header (column names) of the queried table and the question as input to learn a joint representation. The decoder is an LSTM with attention mechanism. There are three output layers corresponding to three decoding types, which restrict the vocabulary it can sample from at each step. The three decoding types are as follows:

- $\tau_V$ (SQL operator): The output has to be a SQL operator, i.e., a terminal from $V = \{$Select, From, Where, Id, Max, Min, Count, Sum, Avg, And, =,

$>$, $<$, <END>, <GO>$\}$.
- $\tau_C$ (column name): The output has to be a column name, which will be copied from either the table header or the query section of the input sequence. Note that the column required for the correct SQL output may or may not be mentioned explicitly in the question.
- $\tau_Q$ (constant value): The output is a constant to be copied from the question section of the input sequence.

The grammar of SQL expressions in the the WikiSQL dataset can be described in a regex form as "Select $f$ $c$ From $t$ Where $(c$ $op$ $v)^*$" ($f$ refers to an aggregation function, $c$ refers to a column name, $t$ refers to the table name, $op$ refers to an comparator and $v$ refers to a value). This can be represented by a decoding-type sequence $\tau_V \tau_V \tau_C \tau_V \tau_C \tau_V (\tau_C \tau_V \tau_Q)^*$, which ensures that only decoding-type corrected tokens can be sampled at each decoding step.

#### 2.1.1. TRAINING

The model is trained from question-SQL program pairs $(X, Y)$, where $Y = [y^{(1)}, \dots, y^{(|Y|)}]$ is a sequence representing the ground truth program for question $X$. Different typed decoder cells are trained with different loss functions.

$\tau_V$ **loss:** This is the standard RNN case, i.e. the loss for an output token is the cross-entropy of the one-hot encoding of the target token and the distribution over the decoder vocabulary $\mathcal{V}$:

$$loss_\mathcal{V}(k) = -\operatorname{onehot}(y^{(k)}) \cdot \log(\operatorname{softmax}(W_\mathcal{V}(\alpha_\mathcal{V}^{(k)} O_e) + b_\mathcal{V}))$$

where $W_\mathcal{V}, b_\mathcal{V}$ are trainable variables, and $\alpha_\mathcal{V}^{(k)} O_e$ denotes attention over an embedding $O_e$ of the input sequence $X$.

$\tau_C$, $\tau_Q$ **loss:** In this case, the objective is to copy a correct token from the input into the output. As the original input-output pair does not explicitly contain any pointers, we first need to find an index $\lambda_k \in [1, \dots, |X|]$ such that $y^{(k)} = x^{(\lambda_k)}$. In practice, there are often multiple such indices, i.e., the target token appears several times in the input query (e.g., both as a column name supplied from the table information and as part of the user question). To express this, we use the "Sum-Transfer" loss, described below.

The probability of copying a token $v$ in the input vocabulary is the sum of probabilities of pointers that point to the token $v$:

$$\phi_{\text{sum}}^{(k)}(v) = \sum_{1 \le l \le |X|} \{\alpha^{(k,\ell)} \mid x^{(l)} = v\}$$

where $\alpha^{(k,\ell)}$ denotes the attention over input embeddings. Based on this, the Sum-Transfer loss is defined as:

$$loss_C^{\text{val}}(k) = -\operatorname{onehot}(y^{(t)}) \cdot \log([\phi^{(k)}(v) \mid v \in \operatorname{Set}(X)]).$$

When training with the Sum-Transfer loss function, we adapt the outputs of the $\tau_Q$ and $\tau_C$ decoder cells to be the tokens with the highest transferred probabilities, computed by $\mathrm{argmax}_{v \in X}(\phi_{\mathrm{sum}}^{(k)}(v))$, so that decoding results are consistent with the training goal.

The overall loss for a target output sequence can then be computed as the sum of the appropriate loss functions for each individual output token.

### 2.2. Execution-Guided Decoding

Since SQL programs are executable, we can use the SQL semantics to guide the repairing of decoded programs (or partial programs) that throw errors during execution. We consider the following two types of errors that could be identified by the execution engine:

- *Runtime error*: A program $p$ throws a run-time error if it has a component whose operator type mismatches its operands type. Such an error could be caused by the mismatch between the aggregation function and the target column (e.g., sum over a column with string type) or the mismatch between condition operator and its operands (e.g., applying $>$ to a column of float type and a constant of string type).

- *Empty output*: When executed, a program $p$ could return a empty result if the predicate generated by the decoder is overly restricted (e.g., a predicate $c = v$ is generated but the constant $v$ in a predicate is not in a column $c$).

In either case, executing the decoded program cannot yield a valid answer to the user's question. To repair erroneous programs, we propose the following two repair approaches.

**Ensemble Approach** We first train $k$ models $(M_1, \ldots, M_k)$ with different random seeds and then use ensembling to repair erroneous programs. When the model $M_i$ returns an erroneous program $p_i$, we invoke the model $M_{i+1}$ to regenerate a new program $p_{i+1}$, until we find an error-free program or finish querying all $k$ models.

**Local Repair Approach** Unlike the ensemble approach that requires the repair process to regenerate new programs from multiple models, the local repair approach repairs the program on-the-fly by leveraging evaluation results of *partial programs*. After decoding the aggregation operator $f$, the aggregation column $c$ and the table $t$, we run the execution engine over the partial program "Select $f$ $c$ From $t$ Where True" to determine whether $f$ and $c$ are compatible. If not, we re-generate $f', c'$ from the set of compatible $(f, c)$ pairs with highest joint probability according to the token distribution produced by the decoder; and then proceed to the decoding of predicates. Similarly,

when decoding a predicate $c_1\ op\ c_2$, we evaluate the partial program with the predicate to check whether the predicate triggers a type error or results in an empty output; if so, we compute the predicate $c_1'\ op'\ c_2'$ with the highest joint probability from the set of error-free predicates $(c_1, op, c_2)$. In practice, instead of computing all possible alternative local repairs, we parameterize the approach with a parameter $k$ to restrict the number of alternative tokens considered at each decoding step (i.e., beam size $k$ for each decoder cell). This approach resembles a beam decoder; however, instead of generating the top-$k$ highest probability programs, the local repair approach utilizes the evaluation result of the partial program to guide the search to avoid decoding erroneous programs.

As we will show later, these two approaches are capable of repairing different types of errors and are both effective in improving decoding accuracy.

## 3. Evaluation

We evaluate our model on WikiSQL dataset (Zhong et al., 2017) by comparing it with prior work and our model with different sub-components to analyze their contributions.

### 3.1. Experiment Setup

We use the sequence version of the WikiSQL dataset with the default train/dev/test split. Besides question-program pairs, we also use the tables in the dataset to preprocess the dataset. The dataset contains 56,324 training pairs, 8,421 dev pairs, and 15,878 test pairs. The detailed data preprocessing, column annotation, and model setups are described in Appendices A.1.

In the execution-guided repair phase, we consider two instances of the ensemble repair approach (one with an ensemble size 3 and one with ensemble size 6) as well as two instances of the repair model (one with beam size 3 and one with beam size 5).

### 3.2. Overall Result

Table 1 shows the results of our model compared against the original Seq2SQL baseline (Zhong et al., 2017) as well as two most recent state-of-the-art models: TypeSQL (Yu et al., 2018) and Coarse2Fine (Dong & Lapata, 2018a). We report both the accuracy computed with exact syntax match ($\mathrm{Acc}_{\mathrm{syn}}$) and the accuracy based on program execution result ($\mathrm{Acc}_{\mathrm{ex}}$). The execution accuracy is higher than syntax accuracy as syntactically different programs can generate same results (e.g., programs only differ in predicate orders).

The comparison result shows that while our base model does not achieve a high execution accuracy compared to the two state-of-the-art models, the repair process can effec-

tively boost the base model accuracy to achieve a similar accuracy as the Coarse2Fine model (78.3% v.s. 78.5%). In particular, since the repair approach is orthogonal to the underlying base model implementation, it can also be applied to improve other base models such as Coarse2Fine itself.

| Model | Dev | | Test | |
|-------|-----|-----|------|-----|
| | $Acc_{syn}$ | $Acc_{ex}$ | $Acc_{syn}$ | $Acc_{ex}$ |
| Seq2SQL (2017) | 49.5 | 60.8 | 48.3 | 59.4 |
| TypeSQL (2018)[1] | – | 74.5 | – | 73.5 |
| Coarse2Fine (2018a) | 72.5 | 79.0 | 71.7 | 78.5 |
| Our Base Model | 61.8 | 72.5 | 62.3 | 71.9 |
| Base + EG Ensemble (3) | 66.6 | 77.3 | 66.7 | 76.9 |
| Base + EG Ensemble (6) | 67.5 | 78.4 | 67.7 | 78.1 |
| Base + EG Local Repair (3) | 65.8 | 77.9 | 66.1 | 77.6 |
| Base + EG Local Repair (5) | 66.2 | 78.5 | 67.9 | 78.3 |

Table 1: Dev and test accuracy (%) of the models, where $Acc_{syn}$ refers to syntax accuracy and $Acc_{ex}$ refers to execution accuracy. "+ Ensemble ($k$)" indicates that model outputs are repaired using an ensemble of $k$ models, and "+ EG Local Repair ($k$)" indicates that model outputs are repaired using the local repair strategy with beam size $k$.

### 3.3. Repair Model

Table 2 shows the number of erroneous programs generated by each model. The result shows that both repair approaches can effectively reduce the number of erroneous programs.

| Model | Dev | Test |
|-------|-----|------|
| Our Base Model | 1348 | 2550 |
| Base + EG Ensemble (3) | 519 | 1063 |
| Base + EG Ensemble (6) | 304 | 696 |
| Base + EG Local Repair (3) | 196 | 379 |
| Base + EG Local Repair (5) | 109 | 217 |

Table 2: The number of erroneous programs generated by different models.

Table 3 shows how the two repair approaches differ in their performances with respect to program size. While both approaches can significantly improve execution accuracies of programs by the base model, the ensemble approach tends to perform better in repairing programs of larger sizes while the local repair approach performs better in shorter programs. This difference is mainly caused by the fact that the local repair approach only repairs program errors in component level and lacks the ability to track full program correctness (as different predicates may not be consistent with each other after repairs). Instead, the ensemble model approach keeps whole program consistency, but the size of ensemble model limits the number of alternative programs that can be used in repairing the original decoding result. The two approaches can potentially be combined to further improve decoder performance.

---

[1] TypeSQL model generates programs in canonical forms and $Acc_{syn}$ does not apply to the model.

| Model | Ground truth predicate size | | | | |
|-------|-----|-----|-----|-----|-----|
| | 0 | 1 | 2 | 3 | 4 |
| Base Model | 57.8 | 77.6 | 63.0 | 57.5 | 42.4 |
| Base + EG Ensemble (6) | 68.8 | 80.5 | 76.3 | 66.1 | 63.6 |
| Base + EG Local Repair (5) | 71.9 | 82.6 | 71.4 | 64.8 | 51.5 |

Table 3: Breakdown results showing the relationship of program size and the execution accuracy (%). Program sizes are measured by the number of predicates in ground truth.

## 4. Related Work

*Semantic Parsing.* Nearest to our work, mapping natural language to logic forms has been extensively studied in natural language processing research (Zettlemoyer & Collins, 2012; Artzi & Zettlemoyer, 2011; Berant et al., 2013; Wang et al., 2015; Iyer et al., 2017; Iyyer et al., 2017). Dong & Lapata (2016); Alvarez-Melis & Jaakkola (2017); Krishnamurthy et al. (2017); Yin & Neubig (2017); Rabinovich et al. (2017); Xu et al. (2017); Dong & Lapata (2018a) are closely related neural semantic parsers adopting tree-based decoding or canonical grammar decoding that also utilize grammar production rules as decoding constraints. Our base model foregoes the complexity of generating a full parse tree and never produces non-terminal nodes. Instead, it retains the simplicity and efficiency of a sequence decoder. Furthermore, the use of the executable semantics of generated programs to guide repairing program compensates the simplicity of the base model. As the repair approach is orthogonal to base model design, it can potentially be combined to boost the performance of other base models.

*Orthogonal Approaches.* Entity linking (Calixto et al., 2017; Yih et al., 2015; Krishnamurthy et al., 2017; Yu et al., 2018) is a technique used to link knowledge between the encoding sequence and knowledge base (e.g., table, document) orthogonal to the neural encoder decoder model. This technique can potentially be used to address our limitation in our deterministic column annotation process.

## 5. Conclusion

We presented a new sequence-to-sequence based neural architecture to translate natural language questions over tables into executable SQL queries. Our approach uses a simple type system to guide the decoder to either copy a token from the input using a pointer-based copying mechanism or generate a token from a finite vocabulary. It uses a sum-transfer value based loss function that transforms a distribution over pointer locations into a distribution over token values in the input to efficiently train the architecture. We propose two model-independent approaches, an ensemble based approach and a local repair approach, with program execution-based guidance to effectively eliminate programs that cause faults or lead to empty results. Our evaluation on the WikiSQL dataset shows that our model achieves close to state-of-the-art results with lesser model complexity.

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

# A. Appendix

## A.1. Experiment Detail

**Data preprocessing** We first preprocess the WikiSQL dataset by running both tables and question-query pairs through Stanford Stanza (Manning et al., 2014) using the script included with the WikiSQL dataset, which normalizes punctuation and cases of the dataset. We further normalize each question based on its corresponding table: for table entries and columns occurring in questions or queries, we normalize their format to be consistent with the table. This process aims to eliminate inconsistencies caused by different whitespace, e.g. for a column named "country (endonym)" in the table, we normalize its occurrences as "country ( endonym )" in the question to "country (endonym)" so that they are consistent with the entity in table. Note that we restrict our normalization to only whitespace, comma (','), period ('.') and word permutations to avoid over-processing. We do not edit tokens: e.g., a phrase "office depot" occurring in a question or a query will not be normalized into "the office depot" even if the latter occurs as a table entry. Similarly, "california district 10th" won't be normalized to "california 10th", and "citv" won't be normalized to "city". We also treat each occurrence of a column name or a table entry in questions as a single word for embedding and copying (instead of copying multiple times for multi-word names/constants).

**Column Annotation** We annotate table entry mentions in the question with their corresponding column name if the table entry mentioned uniquely belongs to one column of the table. The purpose of this annotation is to bridge special column entries and their column information that cannot be learned elsewhere. For example, if an entity "rocco mediate" in the question only appears in the "player" column in the table, we annotate the question by concatenating the column name in front of the entity (resulting in "player rocco mediate"). This process resembles the entity linking technique used by Krishnamurthy et al. (2017); Yu et al. (2018), but in a conservative and deterministic way.

**Model Setup** In our base model, we use the pre-trained $n$-gram embedding by Hashimoto et al. (2017) (100 dimensions) and the GloVe word embedding (100 dimension) by Pennington et al. (2014); each token is embedded into a 200 dimensional vector. Both the encoder and decoder are 3-layer bidirectional LSTM RNNs with hidden states sized 100. The model is trained with question-query pairs with a batch size of 500 for 100 epochs. During training, we clip gradients at 10 and add gradient noise with $\eta = 0.3, \gamma = 0.55$ to stabilize training (Neelakantan et al., 2015). The model is implemented in Tensorflow and trained using the Adagrad optimizer (Duchi et al., 2011).

## A.2. Repair Model Statistics

Table 4 shows how repair models repair different parts of generated programs. We notice that the key improvement comes from repairing of the predicates.

| Model | $Acc_{agg}$ | $Acc_{sel}$ | $Acc_{cond}$ |
|---|---|---|---|
| Base Model | 88.8 | 85.6 | 78.5 |
| Base + EG Ensemble (6) | 88.9 | 85.7 | 86.2 |
| Base + EG Local Repair (5) | 88.9 | 85.5 | 84.8 |

Table 4: Breakdown results on WikiSQL. $Acc_{agg}$, $Acc_{sel}$, and $Acc_{cond}$ are the accuracies (%) of syntactical matches on aggregation function, select column, and condition predicates between the synthesized SQL and the ground truth respectively over the dev set.

## A.3. Examples of generated queries

We show a few examples wrongly generated by our base model that are subsequently repaired.

**Example 1**

- Table: 2-16668557-1 [poll source, sample size, margin of error, date, democrat, republican]

- Question: *What was the date of the poll with a sample size of 496 where republican mike huckabee was chosen?*

- Solution: `Select` date `From` 2-16668557-1 `Where` republican = mike huckabee `And` sample size = 496

- Prediction: `Select` date `From` 2-16668557-1 `Where` sample size = 496 `And` republican = 496

- Ensemble Repair: `Select` date `From` 2-16668557-1 `Where` sample size = 496 `And` republican = mike huckabee

- Local Repair: `Select` date `From` 2-16668557-1 `Where` sample size = 496 `And` republican = mike huckabee

(Remarks: Both repair approaches locate and repairs the wrong predicate republican = 496 in the initial prediction.)

**Example 2**

- Table:1-11336756-6 [route name, direction, termini, junctions, length, population area, remarks]

- Question: *Which population areas have "replaced by us 83" listed in their remarks section ?*

- Solution: `Select` population area `From` 1-11336756-6 `Where` remarks = replaced by us 83

- Prediction: `Select` population area `From` 1-11336756-6 `Where` remarks = remarks

- Ensemble Repair: `Select` route name `From` 1-11336756-6 `Where` remarks = replaced by us 83 `And` remarks = replaced by us 83

- Local Repair: `Select` population area `From` 1-11336756-6 `Where` remarks = replaced by us 83

(Remarks: The ensemble approach make the output program executable but chooses the wrong select column, which still leads to a wrong solution.)

**Example 3**

- Table: 2-17287870-1 [name, built, listed, location, county]

- Question: *What bridge in sheridan county was built in 1915 ?*

- Solution: `Select` name `From` 2-17287870-1 `Where` county = sheridan `And` built = 1915

- Prediction: `Select` county `From` 2-17287870-1 `Where` county = 1915 `And` county = 1915

- Ensemble Repair: `Select` county `From` 2-17287870-1 `Where` built = 1915 `And` county = sheridan

- Local Repair: `Select` county `From` 2-17287870-1 `Where` county = sheridan `And` county = sheridan

(Remarks: In this example, while both of the repair fail to repair the column name in the select clause, the ensemble model successfully repaired the predicate. The local repair approach's repair result is an executable yet incorrect program, as it ignores the second predicate.)