# OpenReview forum: "Execution-Guided Neural Program Decoding"
_ICML.cc/2018/Workshop/NAMPI — NAMPI 2018_

### Review · AnonReviewer2 · 2018-06-21
**Nice paper containing two interesting, orthogonal ideas**

**Rating:** 6
**Confidence:** 3

**Review:**

This paper describes a method for translating natural language queries into SQL queries. They take a reasonably standard sequence-to-sequence RNN model with attention and copying, and add two new ideas. First, because the SQL expressions in their dataset have a fixed regex structure [Select f c from t where (c op v)], and the elements at each place in the structure have a fixed type, they restrict the range of available candidates for each placeholder to elements of the appropriate type. They design separate loss functions for each type: operator, column name, constant value. Second, they introduce a repair mechanism to remove run-time errors from the generated SQL queries. These are both interesting ideas, and they are orthogonal. (The authors say this explicitly in Section 3.2).

The paper is largely well written and very clear. There are a couple of typo's in the abstract, which is slightly off-putting, but after that the paper reads well. The typo's are: " a simple type system of SQL to constrain[t]" on line 15 and "return empty result" on line 20.

The results they show are almost state of the art when using repair. They also say that their local and ensemble repair approaches could be applied to other models as well. It would be very interesting to see how repair helped in these other models.

The authors state that their model achieves close to state-of-the-art results but with less model complexity. Could the authors provide numbers to back up this claim? For example, their decoder vocabulary has a size of 15 because of the type restrictions.

---

### Review · AnonReviewer3 · 2018-06-25
**A very educative and insightful paper**

**Rating:** 9
**Confidence:** 3

**Review:**

In this paper, authors propose a seq2seq neural architecture for translating natural language questions over tables into executable SQL queries. The proposed approach uses a simple typing system for making the decoder either copy a token from the input, or select a token from a vocabulary. Furthermore, for eliminating runtime errors and empty output errors, authors propose two model-independent approaches. Despite its simplicity, the proposed method's results are very close to SOTA, with a significantly lower complexity.

---

### Decision · ~NAMPI_Admin1 · 2018-06-28
**Paper3 Final Decision**

Accept